# Multi-Target Detection of Nuts and Peanuts as Hidden Allergens in Bakery Products through Bottom-Up Proteomics and High-Resolution Mass Spectrometry

**DOI:** 10.3390/foods12040726

**Published:** 2023-02-07

**Authors:** Anna Luparelli, Ilario Losito, Elisabetta De Angelis, Rosa Pilolli, Linda Monaci

**Affiliations:** 1Institute of Sciences of Food Production, National Research Council (ISPA-CNR), Via G. Amendola, 122/O, 70126 Bari, Italy; 2Department of Chemistry, University of Bari “Aldo Moro”, Via E. Orabona 4, 70126 Bari, Italy; 3SMART Inter-Department Research Center, University of Bari “Aldo Moro”, Via E. Orabona 4, 70126 Bari, Italy

**Keywords:** tree nut allergy, peanut allergy, hidden allergens, LC-MS/MS analysis, peptide markers, bakery products

## Abstract

Due to the growing global incidence of allergy to nuts and peanuts, the need for better protection of consumers sensitive to those products is constantly increasing. The best strategy to defend them against adverse immunological reactions still remains the total removal of those products from their diet. However, nuts and peanuts traces can also be hidden in other food products, especially processed ones, such as bakery products, because of cross-contamination occurring during production. Precautionary labelling is often adopted by producers to warn allergic consumers, usually without any evaluation of the actual risk, which would require a careful quantification of nuts/peanuts traces. In this paper, the development of a multi-target method based on liquid chromatography-tandem high resolution mass spectrometry (LC-MS, MS/MS), able to detect traces of five nuts species (almonds, hazelnuts, walnuts, cashews and pistachios) and of peanuts in an in-house incurred bakery product (cookie) through a single analysis is described. Specifically, allergenic proteins of the six ingredients were used as the analytical targets, and the LC-MS responses of selected peptides resulting from their tryptic digestion, after extraction from the bakery product matrix, were exploited for quantification, following a bottom-up approach typical of proteomics. As a result, nuts/peanuts could be detected/quantified down to mg·kg^−1^ levels in the model cookie, thus opening interesting perspectives for the quantification of hidden nuts/peanuts in bakery products and, consequently, for a more rational use of precautionary labelling.

## 1. Introduction

Food allergy can lead to very serious and sometimes even life-threatening consequences, thus resulting in a major risk for sensitive consumers and a potential burden on health care [1,2]. According to recent research, the European health system has been estimated to spend more than 55 billion euros per year to cope with the consequences of food allergy [3,4]. In contrast, remarkable costs related to the management of allergens and to the consequences of food accidents (recalls caused by the accidental presence of food allergens in products or by incorrect labelling) must be borne by the food industry [3]. Since accidental contamination of food is considered the main cause of food allergies [5], the detection of traces of allergenic ingredients hidden in food matrices using reliable and fast analytical methods represents a primary, although challenging, goal.

Nuts and peanuts certainly have a remarkable importance among hidden allergenic ingredients, due to the relative facility of cross contamination involving them, especially when bakery products are considered, and to the threat they pose for allergic consumers. In recent studies, it was reported that peanuts and tree nuts together account for 70 to 90% of deaths due to anaphylaxis [6,7]. However, it is very difficult to obtain reliable estimates of the prevalence of nut and peanut allergy in the population, since data are often subject to overestimation, due to self-diagnosis and/or to the inclusion of oral allergy syndrome/pollen food allergy syndrome (OAS/PFS) in nut allergy definitions [8]. At the same time, the prevalence is highly dependent on the consumption habits and agricultural characteristics of societies [9]. It appears that hazelnut allergy is the most common nut allergy in Europe and that allergies to peanuts, almonds, walnuts and cashews are widespread in the United States [10]. It is estimated that about 1.2–2% of the world’s population suffers from nut sensitivity [11], an incidence stimulated by the increasing knowledge of the beneficial effects that nuts/peanuts may have on human health, due to their content in micronutrients (vitamins and minerals), and various phytochemical compounds, such as phenolic acids and flavonoids [12].

The allergenic potential of nuts and peanuts is mainly related to their protein profile, which has been studied for decades and is still explored due to its complexity. Proteins involved in the adverse immunological response in susceptible or allergic individuals belong to protein families of 2S albumins, vicilins, legumins, and nsLTPs [13]. The main allergenic proteins identified so far for the species considered in this study are as follows: almond *(Prunus dulcis)*: proteins from Pru du 3 to Pru du 6, Pru du 8, Pru du 10; hazelnut *(Corylus avellana)*: Cor a 1, Cor a 2, Cor a 6, and all proteins from Cor a 8 to Cor a 15; walnut *(Juglans regia)*: from Jug r 1 to Jug r 8; cashew *(Anacardium occidentale)*: from Ana o 1 to Ana o 3; pistachio *(Pistacia vera)*: from Pis v 1 to Pis v 5, and peanut *(Arachis hypogaea)*: from Ara h 1 to Ara h 18 [14,15,16,17,18,19,20]. The allergenic proteins of nuts/peanuts are usually characterized by the resistance to denaturation and proteolysis [21]; moreover, some of them belong to the groups of pathogenesis-related (PR) proteins, profilins, and lipid transfer proteins (LTPs), which are often called panallergens, since they also contribute to the allergenicity of a large group of seeds, pollen, fruits, and other plants [22].

To protect the health and safety of consumers, allergens present in food products must be declared in the list of ingredients. In particular, the EU regulation 1169/2011 requires the declaration on the label of 14 classes of allergens, and related products, including nuts and peanuts, when they are used as ingredients [23]. In contrast, the extreme variability of individual sensitivity to allergens [24] is one of the causes for the absence of a regulatory framework for the management of hidden allergens, which, consequently, has prompted the food industry to make excessive use of Precautionary Allergen Labelling (PAL), leading to a loss of consumer confidence and an underestimation of the risk related to the presence of the allergens declared in the food purchased [25,26]. In order to overcome these drawbacks, threshold levels (like the so-called No Observed Adverse Effect Level (NOAEL) and Lowest Observed Adverse Effect Level (LOAEL), proposed as a part of the VITAL^®^ (Voluntary Incidental Trace Allergen Labelling) program by Australia and New Zealand [27], have been introduced but, at the same time, reliable analytical methods should be developed to detect hidden allergenic ingredients occurring at the levels typically included in the program (vide infra).

When processed foods are involved, the effect of heat treatments on the hidden ingredient allergenicity, especially if related to proteins, is expected to complicate the assessment, and nuts/peanuts are not exceptions [28,29]. Studies conducted on walnuts, for example, have shown an increased digestibility and absorption of proteins following heat treatments, leading to a remarkable influence on the possibility of an allergenic effect [30]. Moreover, the formation of neoantigens (which can increase the allergenicity of a particular nut species) upon thermal treatment cannot be excluded, with the Maillard reaction between an allergenic protein and a carbohydrate being one of the processes involved [31]. It is thus not surprising that several studies have been dedicated to determining the effects of technological and heat treatments on the allergenicity of proteins belonging to nuts, like hazelnut [32,33], cashews and pistachios [34,35], almonds and walnuts [36], and to peanuts [37].

Despite the risk of alterations potentially posed by food processing, proteins represent the best target when analytical methods able to detect and quantify traces of nuts and peanuts as hidden allergens in food products must be developed. Immunochemical and molecular biology-based methods have been proposed among them, most relying, on the Enzyme-Linked Immuno Sorbent Assay (ELISA), Lateral Flow ImmunoAssay (LFIA) and DNA-targeting techniques [38,39,40,41,42]. However, apart from drawbacks like cross-reactivity or hook effects, and degradation of DNA upon food processing, according to the case, such methods lack the ability to detect and quantify different allergenic ingredients via a single analysis. Mass spectrometry (MS)-based methods have thus emerged as very powerful alternatives with inherent multi-allergen capabilities [43,44,45,46]. As emphasized in a recent monography [14], most MS-based methods aimed at the analysis of hidden nuts and peanuts traces in food matrices through the detection of their allergenic proteins, exploit the consolidated bottom-up approach of proteomics. Indeed, allergenic proteins are extracted from food matrices and then in-vitro digested (usually by trypsin), before the resulting peptides are analyzed using liquid chromatography-tandem mass spectrometry. In this case, specific tryptic peptides arising from allergenic proteins can be adopted as quantitative markers of the whole allergenic ingredients when appropriate conversion factors are available. As emphasized, more than 150 different tryptic peptides, arising from major allergenic proteins, have been proposed in the literature as markers for tracing the five nuts herein investigated in a wide range of food products, including chocolate, ice creams, breakfast cereals, and bakery products like bread and cookies [47,48]. Interestingly, the limits of quantifications down to a few mg·kg^−1^ of allergenic ingredient per food product (corresponding to parts-per-million, ppm) were often reported, using both high-resolution and low-resolution mass spectrometers, usually having tandem MS capabilities (like quadrupole-Orbitrap, triple quadrupole, or linear ion trap spectrometers).

Starting from these premises, an analytical method relying on the coupling between liquid chromatography and high-resolution single/tandem mass spectrometry has been developed in our laboratory to detect and quantify, in the same run, traces of five types of nuts (almond, hazelnut, pistachio, walnut and cashew) and of peanuts in a model bakery product using selected tryptic peptides originated from their allergenic proteins. Experiments were performed on a cookie model food produced in-house after carefully adding wheat flour with calculated amounts of previously roasted and powdered nuts/peanuts. A high resolution/accuracy quadrupole-Orbitrap hybrid mass spectrometer, equipped with an electrospray interface and a high energy collisional cell (HCD), was used for peptide detection. The following pipeline was used for the method development: (1) selection of the most appropriate marker peptides for detecting nuts and peanut allergens fulfilling the specific criteria previously reported in literature [48] by analyzing both raw and roasted ingredients, (2) evaluation of the processing effect by searching for the selected peptides in the incurred cookie, (3) evaluation of the most relevant method parameters (linearity, limit of detection (LOD), limit of quantification (LOQ), and intra-day and inter-day repeatability), and (4) comparison of the developed method with the specific VITAL^®^ 3.0 thresholds for applicability purposes.

## 2. Materials and Methods

### 2.1. Chemicals, Materials, and Allergenic Ingredients

The following solvents and chemical compounds: acetonitrile, methanol and water (LC-MS grade), formic acid, ammonium bicarbonate (AB), hydrochloric acid, iodoacetamide (IAA), dithiothreitol (DTT) and tris (hydroxymethyl) aminomethane (Tris) were purchased from Sigma-Aldrich (Milan, Italy). Trypsin Gold Mass Spectrometry Grade was purchased from Promega (Milan, Italy).

Cellulose acetate syringe filters with 5 μm porosity and 25 mm diameter and regenerated cellulose syringe filters with 0.45 μm porosity and 4 mm diameter were purchased from Sartorius (Stedim Biotech GmbH, Goettingen, Germany). Disposable desalting cartridges PD-10 were purchased from GE Healthcare Life Sciences (Milan, Italy). Strata-X (33 μm; 30 mg; 1 mL) SPE cartridges and SepPAk C18 (1 cc, 50 mg) cartridges were purchased, respectively, from Phenomenex (Macclesfield, Cheshire, UK) and Waters (Milan, Italy). The following allergenic ingredients: peanuts (*Arachis hypogaea*), hazelnuts (*Corylus avellana*), pistachios (*Pistacia vera*), almonds (*Prunus dulcis*), cashews (*Anacardium occidentale*) and walnuts (*Juglans regia*) were kindly donated, as raw and roasted products, by Besana S.p.A. (San Gennaro Vesuviano, Naples, Italy).

### 2.2. Isotopically Labelled Peptides

Isotopically labelled synthetic peptides reported in Table 1, referred to allergenic proteins of peanuts, hazelnuts and almonds and purchased from Thermo Fisher Scientific (HeavyPeptide AQUA, Waltham, MA, USA), were used as internal standards during the quantification of peptides related to allergenic ingredients. Specifically, isotopic labelling was performed on the terminal Lysine (K) or on the terminal Arginine (R) of each amino acid sequence, to create a mass increase of +4 and +5 mass units, respectively, compared to the unlabeled peptide. The labelled peptides were singly provided in lyophilized form, to be then resuspended in 5% (*v*/*v*) acetonitrile/water to get the final concentration of 6250 fmol/µL. Peptide solutions were then divided in different aliquots and stored at −20 °C until use.

### 2.3. Incurred and Allergen-Free Cookies Production

In order to evaluate the stability of the potential marker peptides selected for the six allergenic ingredients and to evaluate the impact of the technological treatments, a cookie was produced in-house by adding the six allergens during the dough preparation and before cooking. The following recipe was used for the preparation: 402.4 g of flour, 1 g of salt, 2 g of bicarbonate, 180 g of sugar, 90 g of extra virgin olive oil, 160 g of water, 6.01 g of egg powder, 6.01 g of skimmed milk powder, 6.02 g of roasted peanuts, 6.16 g of roasted hazelnuts, 6.02 g of roasted pistachio, 6.02 g of roasted almonds, 6.16 g of roasted cashews and 6.24 g of roasted walnuts. The dough was divided into discs with a diameter of about 5.5 cm and a thickness of about 1 cm and baked in the oven for 20 min at a temperature of 200 °C. At the end of cooking, the concentration of each individual allergenic ingredient was recalculated considering the loss in water due to heat treatment: the final concentration of each allergenic ingredient corresponded to 7677.64 mg·kg^−1^ of cookie.

Following the same procedure, a sample of nut/peanut-free cookie was produced by replacing allergenic ingredients with flour. Both blank and incurred cookies were finely milled in a blender at 8000 g (Sterilmixer 12 model 6805-50, PBI International) by iterations of four cycles of blending (30 s ON and 10 s OFF to prevent overheating of the material). Ground blank and incurred cookies were sieved through a 1 mm sieve, spread on a large tray (50 cm × 50 cm) and manually mixed for homogeneity. A simulated contamination at a 1000 mg·kg^−1^ level was subsequently obtained by mixing appropriate quantities of powdered allergen-free and incurred cookies; lower levels of contamination were obtained by mixing the purified protein extract obtained from 1000 mg·kg^−1^ incurred cookie with the nut/peanut-free cookie protein extract. Allergen concentrations were defined as milligram of allergenic ingredient per kilogram of matrix (mg_ingr_/kg), unless otherwise specified.

### 2.4. Sample Preparation

#### 2.4.1. Extraction and Tryptic Digestion of Proteins from Raw and Roasted Allergenic Ingredients

Raw and roasted allergenic ingredients (almonds, hazelnuts, walnuts, pistachios, cashews and peanuts) were subjected to the following sample preparation protocol. First, the raw and roasted samples were ground with Sterilmixer 12 model 6805-50 (PBI International, Milan, Italy), carrying out in 5 steps of 15 s at speed 8. Proteins were subsequently extracted by the addition of 10 mL of Tris-HCl 200 mM buffer at pH 9.2 with 5 M urea to 2 g of each sample, followed by stirring in an orbital shaker for 30 min. At the end, the suspensions were kept for 15 min in an ultrasound bath and finally subjected to centrifugation at 4000 rpm for 10 min. Subsequently, the supernatant taken from each sample was filtered on cellulose acetate filters (5 μm) and diluted 1: 5 (*v*/*v*) with a 50 mM solution of NH_4_HCO_3_, in order to reduce the urea concentration to a value (1 M) compatible with the enzymatic activity of trypsin.

Enzymatic digestion (digestion volume: 600 μL per sample) was then performed using the following steps: (i) denaturation at 95 °C for 15 min; (ii) reduction of proteins through the addition of 60 μL of dithiothreitol (DTT) 500 mM, prepared in NH_4_HCO_3_ 50 mM, followed by shaking for 30 min at 500 rpm at 60 °C; (iii) alkylation of the proteins, obtained by adding 120 μL of iodoacetamide (IAA) 100 mM prepared in NH_4_HCO_3_ 50 mM and leaving the mixture for 30 min in the dark at room temperature; (iv) digestion of the extracted proteins, carried out by adding 15 μL of trypsin (concentration 1 μg/μL) to the mixture and leaving the sample for 16 h at 37 °C under stirring (500 rpm). In particular, the reduction step promoted by DTT and the subsequent alkylation with IAA allowed to reduce and finally block the cysteine sulfhydryl groups with alkyl groups, avoiding the formation of disulfide bridges. The digestion reaction was stopped by adding 7 μL of 6 M HCl.

The digestion mixture was then centrifuged at 13,000 rpm for 10 min, and the supernatant was purified by loading on 1 mL SPE columns packed with StrataX stationary phase (30 mg, Phenomenex, Castel Maggiore, Bologna, Italy). The column was firstly conditioned by adding 1 mL of methanol (3 times) and 1 mL of water (3 times), and then, 500 μL of tryptic digest was loaded; a washing step with 1 mL of water (2 times) and with 1 mL of methanol 5% was performed afterwards. The purified peptides were finally eluted from the column by adding 1 mL of a 1:1 (*v*/*v*) acetonitrile/methanol mixture containing 2% formic acid. After elution, the samples were dried under nitrogen flow, resuspended in 500 μL of a 90:10 (*v*/*v*) water/acetonitrile mixture containing 0.1% formic acid and filtered on a 0.45 μm syringe filter. Samples were stored at −20 °C until the LC-MS analysis was performed or analyzed immediately after peptide purification.

#### 2.4.2. Extraction and Tryptic Digestion of Proteins from Incurred and Allergen-Free Cookies

The workflow adopted for the extraction and tryptic digestion of proteins from incurred or allergen-free cookies was based on previous studies on allergens analysis [43,49,50].

Cookies incurred with allergenic ingredients at a 1000 mg·kg^−1^ level and allergen-free cookies, prepared as described before, were first subjected to protein extraction through the addition of 24 mL buffer Tris-HCl 200 mM at pH 9.2 with 5 M urea to 1.2 g of sample, after grinding and sieving the product with a 1 mm sieve, followed by stirring in an orbital shaker for 30 min. At the end, the suspensions were kept for 15 min in an ultrasound bath and the extract was centrifuged at 4000 rpm for 10 min and filtered on cellulose acetate filters (5 μm) to remove coarse particles. The obtained filtrates were subjected to purification based on Size Exclusion Chromatography (SEC) with PD-10 packed columns conditioned as follows: storage buffer washing with 4 mL of Milli-Q (water 3 times), and exchange with 4 mL of NH_4_HCO_3_ 50 mM (3 times). The columns were then placed in 50 mL Falcon tubes with the appropriate adapters, and the last wash was carried out with 4 mL of ammonium bicarbonate 50 mM. The columns were then subjected to centrifugation at 1000× *g* for 2 min, the waste was discarded and 2.5 mL of crude protein extract from each sample was loaded. Centrifugation at 1000× *g* was performed for 2 min afterwards, and the eluted purified protein extract was recovered. The purified extracts of allergen-free cookies were used to dilute the protein extracts obtained from 1000 mg·kg^−1^ incurred cookies, in order to produce extracts containing proteins related to concentrations or each allergenic ingredient ranging between 10 and 200 mg·kg^−1^. A volume of 1000 μL of each sample was then submitted to the enzymatic digestion phase as previously described (Section 2.4.1).

The digestion mixture was centrifuged at 13,000 rpm for 10 min, and the supernatant filtered through 0.45 μm filters, purified and pre-enriched by loading on SPE SepPAk C18 columns. Before samples loading, the columns were conditioned by adding 1 mL of methanol (3 times) and 1 mL of ammonium bicarbonate 50 mM (3 times); 1000 μL of digests were loaded into the columns and then washed with 800 μL of Milli-Q water + 0.1% of formic acid. The purified peptides were finally eluted from the stationary phase by adding 500 μL (3 times) of a methanol/water 90:10 solution (*v*/*v*) to the column. After elution, the samples were dried under nitrogen flow and then resuspended with 100 μL of a water/acetonitrile 95:5 (*v*/*v*) mixture containing 0.1% formic acid.

Isotopically labelled peptides described in Section 2.2, used as internal standards, were finally added to the purified digests, each at a final concentration of 625 fmol/μL.

Samples were stored at −20 °C until the LC-MS analysis was performed or analyzed immediately after peptide purification.

### 2.5. Liquid Chromatography—Mass Spectrometry: Instrumentation and Conditions

The same conditions were adopted for the LC-MS and MS/MS analyses of the tryptic digests of proteins extracted from raw or roasted allergenic ingredients and from cookies incurred with them. The analyses were performed on a platform including an Ultimate 3000 liquid chromatograph coupled to a quadrupole-Orbitrap high-resolution hybrid mass spectrometer equipped with a heated ElectroSpray Ionization (HESI) interface and a Higher Collisional energy Dissociation (HCD) cell, for fragmentation of precursor ions (*Q-Exactive Plus*, Thermo Fisher Scientific, Waltham, MA, USA). Tryptic peptide mixtures were separated using reversed phase liquid chromatography based on an Aeris Peptide column (150 × 2.1 mm, packed with 3.6 µm particles and characterized by a XB-C18 stationary phase), purchased from Phenomenex (Castel Maggiore, Bologna, Italy).

Two solvents were used for chromatographic separation of tryptic peptides, based on gradient elution: water + 0.1% formic acid (A) and acetonitrile + 0.1% formic acid (B). The chromatographic gradient was the following: from 0 to 35 min, B was increased from 5 to 35%; from 35 to 36 min, B was increased from 35 to 90%; in the following 10 minutes (36–46 min) B was kept isocratic at 90%; from 46 to 47 min, B was returned to 5% and then kept constant for other 20 min to guarantee column reconditioning. Column temperature was maintained constant at 30 °C along the entire chromatographic run (67 min); the flow rate was set to 200 μL/min while volume injection was 20 μL.

MS and MS/MS analyses were carried out in positive polarity using two acquisition modes; the first was a Full-MS/dd-MS^2^ one, implying the alternated acquisition of MS spectra and of MS/MS spectra in the data-dependent (dd) mode. In this case, the Full-MS event was based on the following settings: microscan 1, resolution 70 k, Automatic Gain Control (AGC) target 10^6^, maximum injection time 30 ms and scan range 200–2000 *m*/*z*. The dd (data-dependent)-MS^2^ event was based on the following main settings: microscan 1, resolution 17.5 k, AGC target 10^5^, maximum injection time 60 ms, loop count 5, TopN 5, isolation window 2.0 *m*/*z*, isolation offset 0.4 *m*/*z* and stepped collision energy 27, 30. The second acquisition mode was a Full-MS/AIF (All Ion Fragmentation) one: in this case, the Full-MS events were alternated with the fragmentation of all ions generated in the HESI source. The following parameters were set for the Full MS event: microscan 1, resolution 140 k, AGC target 10^6^, maximum injection time 200 ms and scan range 300–2000 *m*/*z*. As for the AIF event, the following parameters were adopted: microscan 1, resolution 70 k, AGC target 10^6^, maximum injection time 200 ms, (N) CE/stepped nce: 27.30 and scan range 250–2000 *m*/*z*.

The following HESI interface and ion optics parameters were set as described: Spray Voltage 3,4 kV Capillary Temperature ) 320 °C; Sheath Gas 25 (a.u.);Auxiliary gas flow rate 11 (a.u.) and S-Lens RF Level 55.

### 2.6. Identification of Allergenic Proteins and Peptides Using Bioinformatics

Raw data obtained by the LC-MS and MS/MS analysis of the protein digests referring to raw and roasted nuts/peanuts ingredients and incurred cookies were finally processed via software to identify candidate markers peptides for allergenic ingredients detection, along with the proteins they belonged to. The Proteome Discoverer™ 2.1 software (Thermo Fisher Scientific, Waltham, MA, USA), based on the Sequest HT algorithm, was used for this purpose. Specifically, a customized database including all the protein entries related to Prunus Dulcis (ID: 3755, 53,241 sequences), Corylus Avellana (ID: 13451, 492 sequences), Arachis Hypogaea (ID: 3818, 101,959 sequences), Juglans regia (ID: 51240, 45,763 sequences), Anacardium occidentale (ID: 171929, 97 sequences) and Pistacia vera (ID: 55513, 106 sequences) was used for Sequest HT search. The following parameters were set for proteins/peptides identification: specific cleavage: trypsin; tolerance on precursor and product ions *m*/*z* values: 5 ppm and 0.05 Da, respectively; peptide length: 5–144 amino acids; static modification: cysteine carbamidomethylation; and dynamic modification: methionine-oxidation, glutamine/asparagine-deamidation, *N*-terminal glutamine cyclization to pyroglutamate, *N*-terminal protein acetylation. Only trustful peptide-spectrum matches were accepted (matching of at least three consecutive product ions of y- or b- series, total ion current of MS/MS spectra > 500), and a minimum of two peptides was set as the threshold for protein identification, after filtering the peptide list to the sequences assigned with at least medium confidence (False Discovery Rate—FDR < 5%) and a Score Sequest HT ≥ 1.

As for the final selection of marker peptides, only sequences fulfilling the following criteria were considered: (1) 7 to 20 amino acids, (2) not including amino acids C and M and the combination NG and 3) missed cleavage = 0.

### 2.7. Evaluation of the Performance of the Developed Method

#### 2.7.1. Sensitivity: Calibration and Calculation of LOD and LOQ Values

Starting from the protein extracts of 1000 mg·kg^−1^ incurred cookie and of the allergen-free cookie, extracts at different contamination levels, namely 10, 20, 50, 100 and 200 mg·kg^−1^, were prepared for calibration purposes. The extracts were subsequently subjected to tryptic digestion, and a mixture of labelled peptides (Table 1), used as internal standards, was finally added to the purified digests.

Peak area values obtained from eXtracted Ion Current (XIC) chromatograms referred to the doubly charged ion of each selected peptide at each contamination level were normalized to the peak areas obtained for the isotopically labelled peptides for the corresponding ingredient. In order to enhance the extraction selectivity, *m*/*z* ranges with a 0.005 width were always adopted for ion current extraction. For peptides belonging to nuts for which no isotopically labelled peptides were available (pistachio, cashew and walnut), the normalization of the respective peak area was obtained by taking into account the closest one, in terms of mass and amino acids composition, among the peptides listed in Table 1. 

Method sensitivity was evaluated in incurred cookie samples by the calculation of both limit of detection (LOD) and quantification (LOQ). The LODs and LOQs were calculated as 3 and 10 times the standard deviation on the intercept of the calibration line divided by the slope of the calibration curve, respectively.

#### 2.7.2. Evaluation of the LC-MS Analysis Precision

To evaluate the precision of the LC-MS analysis of tryptic digests, a dedicated experiment was carried out by preparing a protein extract corresponding to a cookie incurred with 50 mg·kg^−1^ of each allergenic ingredient, which was subsequently digested according to the procedure described before. Five LC-MS analysis replicates were performed during the same day on the same protein extract, for three consecutive days, to estimate the intra-day and inter-day precision, that was evaluated as the percentual Coefficient of Variation (CV%) observed for peak areas retrieved from extracted ion current chromatograms obtained for selected marker peptides.

## 3. Results and Discussion

### 3.1. Protein Identification and Marker Peptide Selection for Raw and Roasted Allergenic Ingredients (Almond, Walnut, Cashew, Hazelnut, Pistachio and Peanut)

Heat treatments are widely used in the nut processing industry to improve the sensory properties of nuts and ensure their safety. These treatments can change the biochemical properties of nut proteins or produce other substances through chemical reactions within the components of the food matrix, thus potentially affecting the final allergenicity [51]. Some processing methods, such as roasting, may affect solubility, as reported in a recent study on cashew proteins, showing that the content of soluble proteins gradually decreases with prolonged roasting time [52]. In fact, because of roasting, the stable form of a protein, in which the hydrophobic groups are oriented inwards and the hydrophilic groups are oriented outwards, could be compromised. Some studies have shown that although the solubility of proteins may improve after heat treatment in some cases, in other circumstances, proteins may aggregate and reduce their solubility or extractability [53,54]. In the light of this knowledge, the first purpose of this investigation was to evaluate the effects of the heat treatment on the nuts and peanut protein pool, to obtain information on their thermal stability and on the chemical modifications that could occur upon roasting. This information was then considered for the selection of the most reliable marker peptides for nuts/peanuts detection in the cookie matrix. For this purpose, the five tree nuts species and peanuts (raw and roasted) were subjected to protein extraction, followed by tryptic digestion of extracts and LC-MS and MS/MS analysis of digests (according to the protocol described in Section 2.4.1). Raw files deriving from the untargeted analysis, performed through Full-MS/dd-MS2 acquisition, were processed via Proteome Discoverer software and the results filtered and manually validated according to the criteria described in Section 2.6 for protein identification. The numbers of proteins identified for each ingredient in raw and roasted nuts/peanuts are summarized in Figure 1a. As observed in the graph, the impact of the thermal treatment was significant in the case of peanuts, halving the number of proteins recognized in the roasted product. In the other cases, the number of detected proteins were comparable, or even identical, yet the numbers of recognized peptides were generally quite lower, as shown in Figure 1b. This finding suggests that the applied heat treatment may have influenced the stability of the original proteins, inducing their degradation or aggregation with the food matrix or altering their structure [55,56], thus compromising their solubility and, therefore, the extractive yield of the protocol adopted. Moreover, heat-triggered modifications were likely responsible for the lack of recognition of several peptides when roasted ingredients were considered. The results shown in Figure 1 are in accordance with those recently reported by Korte et al. in a study on the impact of processing on the detection of marker peptides for allergenic proteins of almond, pistachio, walnut, cashew, hazelnut and peanut, in which heating was shown to lead to a 20–83% loss of signal, depending on the allergen and on the product type and cooking time [57].

In the perspective of selecting reliable sequences for the quantification of allergenic ingredients, common to raw and roasted ingredients, peptides accounted for in Figure 1b were subsequently subjected to a more severe filtering by excluding those including methionine and/or the asparagine-glycine motif, since the latter are known to undergo oxidation or deamidation, respectively, upon protein processing [48,58]. Additionally, peptides containing cysteine residues that are purposely subjected to carbamidomethylation before proceeding to tryptic digestion were excluded. Finally, two further constraints were adopted, namely, a sequence length between 7 and 20 amino acids and the absence of missed cleavages (i.e., of arginine or lysine residues not located at the carboxylic terminus). When peptides complying with the described criteria and common to raw and roasted ingredients were considered, the list reported in Table 2 was obtained, including 22 common peptides for peanut, 29 for walnut, 28 for pistachio, 10 for cashew, 13 for hazelnut and 19 for almond. This list was the starting point for the selection of peptides to be used as markers of allergenic ingredients in the incurred cookie.

### 3.2. Selection of Marker Peptides for Nuts and Peanuts Quantification in Incurred Cookies

A cookie extract corresponding to a 200 mg·kg^−1^ concentration of each tree nut and of peanuts was subjected to protein extraction followed by tryptic digestion and protein digest analysis using LC-ESI-MS and MS/MS in the Full-MS/dd-MS2 mode, with the goal of verifying how many peptides among those previously selected for raw or roasted ingredients could be identified. Raw data were processed according to the same procedure described in Section 3.1. As a result, only the peptides underlined in the first column of Table 2 were reliably recognized, i.e., 7 for peanut, 18 for walnut, 15 for pistachio, 5 for cashew, 6 for hazelnut and 9 for almond. Different causes can be invoked to explain this result; one of them might be the interaction between nut/peanut proteins and cookie matrix components, potentially affecting the allergenic proteins extractability and/or their tryptic digestion yield. The competition for ionization between peptides arising from allergenic ingredients and those generated from other proteins occurring in the cookie matrix, when not perfectly separated using liquid chromatography, might also play a role. Nonetheless, a good number of marker peptides could be finally retrieved for each ingredient.

In a subsequent step, the response obtained for the selected peptides was evaluated by considering the area of the chromatographic peak detected in the corresponding XIC chromatogram when LC-ESI-MS data referred to the 200 mg·kg^−1^ incurred cookie extract were processed. Examples of XIC chromatograms, obtained by extracting the ion current in a *m*/*z* window centered on the exact *m*/*z* ratio of the doubly charged peptide ion of interest and having a 0.005 width, are shown in Figure 2 for the TANDLNLLILR peptide selected for peanuts. As observed in the figure, the presence of interfering peaks, not due to the selected peptides, was quite rare, due to the extreme narrowness of the ion current extraction window, greatly enhancing the specificity of the data processing. This approach was clearly enabled by the very high accuracy available with the employed mass spectrometer. In any case, even when some minor interfering peaks were also detected in XIC traces, the peak referred to a marker peptide of nuts or peanuts could be easily recognized through the retention time alignment with peaks detected in chromatographic traces referred to specific product ions of that peptide, in turn obtained from AIF acquisitions.

XIC chromatograms like those reported in Figure 2 were exploited for the selection of the three marker peptides providing the highest responses for each ingredient, corresponding to the areas underlying peaks detected in those traces. The sequences of the three peptides finally selected for each ingredient are marked with bold character in the first column of Table 2. The selection of peptides providing the most intense responses aimed at reducing the risk of false negatives that would pose a risk to the safety of the allergic consumers [47]. In contrast, the availability of additional information arising from MS/MS acquisitions led to minimize the risk of confusing eventual isobaric peptides with the actual marker peptides, a risk that is expected to be significant in a complex matrix and that would lead to false positives. As a further control, the allergen-free cookie was also subjected to protein extraction and digestion and to subsequent LC-ESI-MS and MS/MS analysis of the digest in the same conditions adopted for the 200 mg·kg^−1^ incurred cookie, and no significant peaks were detected in the XIC chromatograms at the same retention times of the selected marker peptides.

The possibility that the selected peptide sequences could be found also in other food proteins, thus leading to false positives if those proteins occurred in the analyzed food products, was also evaluated, using the MS-Homology program, available among programs included in the Protein Prospector 6.4.2 portal developed by the University of California at San Francisco (freely accessible at the Internet address: https://prospector.ucsf.edu/prospector/cgi-bin/msform.cgi?form=mshomology (accessed on 2 September 2022)).

Specifically, each peptide sequence was searched against protein sequences stored in the following databases: NCBInr.2013.6.17, SwissProt.2021.06.18 and UniProtKB.2020.09.02. Almost all the selected peptides showed 100% specificity for nut or peanuts proteins, with just a few exceptions: in the case of almond, peptide LLSATSPPR was also found in a predicted protein of *Fibroporia radiculosa* (a brown rot fungus) and in the hypothetical protein PRUPE_ppa004418mg of *Prunus persica* (peach), whereas peptide TEENAFINTLAGR was found in another protein of *Prunus persica* (PRUPE_ppa003759mg). Notably, the third peptide selected as a marker for almonds, ADIFSPR, had 100% specificity towards them, thus enabling the reliable identification of almonds as hidden ingredient. Hazelnut peptide INTVNSNTLPVLR was found also in a 11S globulin of *Carpinus fangiana* (Monkey Tail Hornbeam), a tree of the *Betulaceae* family, whereas walnut peptide ADIYTEQAGR was found also in Legumin A of *Morella rubra* (Chinese bayberry) and in a 11S globulin of *Papaverum somniferum* (Opium poppy). Moreover, in the case of hazelnuts and walnuts, the remaining two peptides among those selected as markers had 100% specificity, thus enabling their unequivocal identification in any case.

Starting from the considerations made so far, the three peptides selected for each allergenic ingredient were further evaluated for application as quantitative markers.

### 3.3. Quantification of Nuts and Peanuts in the Incurred Cookie Based on Marker Peptides: Evaluation of Linearity and Sensitivity

In order to evaluate the linearity and the sensitivity enabled by marker peptides, protein extracts referred to the cookie incurred with 10, 20, 50, 100 and 200 mg·kg^−1^ of each allergenic ingredient were subjected to tryptic digestion, followed by LC-ESI-MS and MS/MS analysis of digests that were appropriately spiked with isotopically labelled peptides, as described in Section 2.7.1. XIC chromatograms were retrieved for each peptide; an example of the doubly charged ion of peptide TANDLNLLILR peptide from the Ara 3 peanut protein (precursor *m*/*z* 628.3721) is shown in Figure 2.

Peak areas obtained from chromatograms such as those reported in Figure 2 were normalized by those referred to isotopically labelled peptides, according to the correspondences evidenced in Table 3. As mentioned before, if isotopically labelled peptides for specific marker peptides were not available, the most similar ones, either in terms of length or in sequence, were used. Calibration graphs were obtained for all the marker peptides, based on normalized XIC peak areas. Two replicates of digestion/analysis were considered for each sample. An example of a calibration plot for each allergenic ingredient is shown in Figure 3. As inferred from the figure, the response linearity was generally excellent, with only one correlation coefficient being lower than 0.99 (see Table 4). As also shown in Table 4, LOD/LOQ values lower than 8/27 mg·kg^−1^ were obtained, the only exception being those related to the LLSATSPPR peptide of almond (14/47 mg·kg^−1^).

Starting from these values, an evaluation of the method performance with respect to threshold levels currently reported for nuts and peanuts was undertaken.

### 3.4. Comparison of the Method Performance with the Thresholds for Nuts and Peanuts Reported in the VITAL^®^ Program

The Allergen Bureau’s VITAL^®^ Program is a standardized allergen risk assessment process in the food industry. Understanding the strengths and limitations of different analytical methods helps analysts in selecting the most appropriate method to meet their analytical needs and provide the risk assessors the necessary information to make decisions to manage risk on a public health level. In this regard, the application of the VITAL^®^ program aims to avoid indiscriminate use of precautionary allergen labelling and thereby protecting its value as a risk management tool. Thus, it aims at minimizing the risk while communicating effectively to allergic consumers [27].

The VITAL^®^ platform (https://vital.allergenbureau.net/dashboard/home/ (accessed on 5 December 2022)) is an easily accessible tool for laboratories and companies, enabling the calculation of different threshold levels for allergenic ingredients, useful for food product labelling. VITAL^®^ uses protein reference doses that are based on the total protein of an allergenic food. For direct comparison with the VITAL^®^ protein reference doses in foodstuff, the reporting unit ideally is the total protein of the allergenic foodstuff, and the methods ideally should refer to the total protein measured of the specific allergenic food. For the calculation of the protein content referred to the allergenic ingredient, universally recognized tables are employed for its conversion.

The VITAL^®^ approach is based on the following three main values: the “Reference Amount”, representing the typical maximum portion (expressed in g) consumed for a food; the “Reference Dose”, referring to the mg of protein below which only the most sensitive individuals (between 1% and 5%) in the allergic population can have an adverse reaction; the “Action Levels”, representing threshold levels (mg·kg^−1^ or ppm) of protein concentrations for food labelling, namely, action level 1: precautionary labelling is not required; action level 2: the “may contain” labelling is required; action level 3: the “contains” labelling is required. The VITAL Scientific Expert Panel (VSEP) identified the ED01 doses (defined as the dose of the total allergen protein that is predicted to produce objective symptoms in 1% of the allergic population) that were adopted as the Reference Doses for VITAL 3.0.

The VITAL grid contains concentrations of cross contact allergen proteins, called Action Levels, which determine, when it is appropriate, to use a precautionary allergen labelling statement. The Action Level concentrations are determined using the Reference Dose information (set by the VSEP) in conjunction with the associated Reference Amount. The VSEP identified the ED01 doses that were adopted as the Reference Doses for VITAL 3.0.

In the case of the five nuts and peanuts under study in this work, the reference amount proposed by the VITAL^®^ program (ED01) is between 0.03 and 0.2 mg depending on the nut. Action levels referred to a fixed amount of allergen containing food ingested are recommended and shown in Table 5. In the same table are also reported the reference doses and action levels 1 and 2 proposed by VSEP and referred to a portion size of 50 g (VITAL^®^ 3.0 version).

In general, VITAL^®^ uses reference doses referred to the total protein content of an allergenic food, although only a subset of proteins actually represents a risk for allergic consumers. Consequently, in order to evaluate the performance of the method based on marker peptides of allergenic proteins, a recalculation of LOD and LOQ values was required. A database developed by the Dutch National Institute for Public Health and the Environment, freely accessible on the Internet (https://nevo-online.rivm.nl/Home/En (accessed on 6 December 2022)), was employed to obtain data on the total protein content of the ingredients of interest; as a consequence, new LOD and LOQ values, expressed as mg of total proteins per kg of cookie matrix, were calculated and are reported in Table 6. In the last column of the table, protein levels that would result in a more severe reaction in 1% of the population of allergic individuals (i.e., the ED01 Eliciting Dose), according to the VITAL^®^ program data, are also reported, for the sake of comparison. As apparent, LOD values obtained using the selected marker peptides were often comparable with those levels, except for the one referred to the already mentioned LLSATSPPR peptide of almond. In contrast, only LOQ values obtained using the three peanut marker peptides and the remaining two peptides selected for almond were suitable for a quantification at the thresholds inferred from the VITAL^®^ program data.

About the calculation of different RDs based on either ED01 or ED05 for protecting allergic consumers’ health, the Joint FAO/WHO Expert Consultation working group on Risk Assessment of Food Allergens considered that offering a dual choice would be problematic and confusing in relation to its practical implementation.

As a result, the committee endorsed the conclusions on hazard characterization and welcomed the proposal of a single Reference Dose (RD) per allergen, opting for a simplification process. In the first instance, for most allergens, the actual ED05 values on which the RDs are based were rounded down to a single significant figure on the basis of the size of the confidence intervals. Exceptions were represented by those allergens for which the data were susceptible to a high degree of bias (e.g., cashew, walnut) or where there could be a high degree of uncertainty for the true value of the ED05 due to the limited number of species tested within a food group. In those cases, the ED05 values were rounded down further than the other foods. Because insufficient data existed for almond, pecan and pistachio and the known cross-reactivities and co-existent allergies between pistachio and cashew, and pecan and walnut, a placeholder RD for pecan and pistachio was proposed as low as 1 mg of total protein from the allergenic source (ED05). Moreover, in view of insufficient information for almond, an RD was proposed at 1.0 mg of total protein from this allergenic source in accordance with the lowest RD for tree nuts.

Consensus RD recommendations were issued for codex priority allergens at 1 mg for cashew, pistachio, almond and walnut; 2 mg for peanut; and 3 mg for hazelnut. Grounding on this, the developed method shows to be very promising and matches the sensitivity required for the RD proposed by the FAO/WHO expert consultation working group that is higher than the dose recommended per the VITAL^®^ program.

It is worth noting that LOD values obtained in this work are in general very competitive compared to that obtained with other developed methods [59,60,61,62] that are able to detect tiny amounts of peanuts and other nuts in a processed matrix below the levels recommended by the VITAL^®^ grid and referred to the consumption of 50 g portion. This confirms the good sensitivity achieved by the present method that can respond to the analytical challenges posed by the VITAL^®^ 3.0 threshold values. Nevertheless, further work is required to propose an official method based on mass spectrometry for the quantification of nuts and peanuts traces in bakery products.

### 3.5. Evaluation of the Precision of the Analytical Method

The intra-day and inter-day precision of the LC-MS analysis performed on the tryptic digest of an incurred cookie protein extract, expressed as percentual coefficients of variation in XIC peak areas of marker peptides, CV%, were evaluated to test the analysis repeatability and reproducibility within the same laboratory. Therefore, the tryptic digestion was performed on day 1 on the protein extract corresponding to a cookie incurred with 50 mg·kg^−1^ of each ingredient, and then, five replicates of analysis were performed each day for days 1, 2, and 3. The intra-day and inter-day CV% were then calculated, considering XIC peak areas of the selected peptides, without normalization to isotopically labelled peptides. As shown in Table 7, the intra-day variabilities were quite different for each day, yet the corresponding inter-day ones were usually comparable with them. This outcome suggests that the tryptic digest was stable over at least a three-day period; thus, MS responses due to marker peptides were simply influenced by the natural fluctuation in the instrumental response, due to the absence of normalization with isotopically labelled standards. The control test described in the present section should be usually performed to evaluate if peptides selected as markers of allergenic proteins are sufficiently stable during low temperature storage of tryptic digest to provide accurate final results.

## 4. Conclusions

A multi-target method based on a bottom-up proteomics approach was developed for the quantification of traces of five tree nuts (walnut, pistachio, cashew, hazelnut and almond) and of peanuts in a bakery product. The food product under study was represented by a cookie prepared in-house, with the six allergenic species added to the ingredients before cooking, to simulate a cross-contamination of raw materials occurring in the production plant.

Upon extraction of the protein fraction from the incurred cookie and its tryptic digestion, the whole peptides-digest was submitted to liquid chromatography-high resolution single and tandem mass spectrometry analysis. Starting from a preliminary evaluation based on the tryptic digests of raw or roasted nuts/peanuts, three peptides related to known allergenic proteins of the six species were selected as reliable quantitative markers for each of them. The limits of detection inferred from calibrations performed on those peptides were usually lower than 8 mg of ingredient added per kg of the bakery product. When expressed in terms of mg of total proteins of each allergenic species per kg of product, those limits proved comparable with the threshold levels eliciting a response in 1% of the allergic population after the consumption of a 50 g portion of food product, based on data reported in the VITAL^®^ 3.0 program. In the case of peanuts and almonds, even the limits of quantification achieved using marker peptides were comparable with those thresholds. Although lower detection/quantification limits should be reached to make the method compliant with very low eliciting thresholds, i.e., those referred to less than 1% of allergic patients, these results indicate that the developed method is a promising approach for the multi-target quantification of allergologically relevant amounts of nuts/peanuts occurring in bakery products because of accidental cross-contamination during food production. Despite the costs related to the installation and operation of a complex MS instrumentation like a quadrupole-Orbitrap mass spectrometer, the method might enable a more rapid and reliable control, compared to other approaches, of hidden allergens in samples obtained in industrial contexts, due to its inherent multi-analyte feature. In turn, the method transfer to industrial quality control laboratories might represent a key step towards a more cautious use of precautionary labelling required for nuts and peanuts species. This represents an important step towards a more cautious use of precautionary labelling required for those nuts and peanuts species.

## Figures and Tables

**Figure 1 foods-12-00726-f001:**
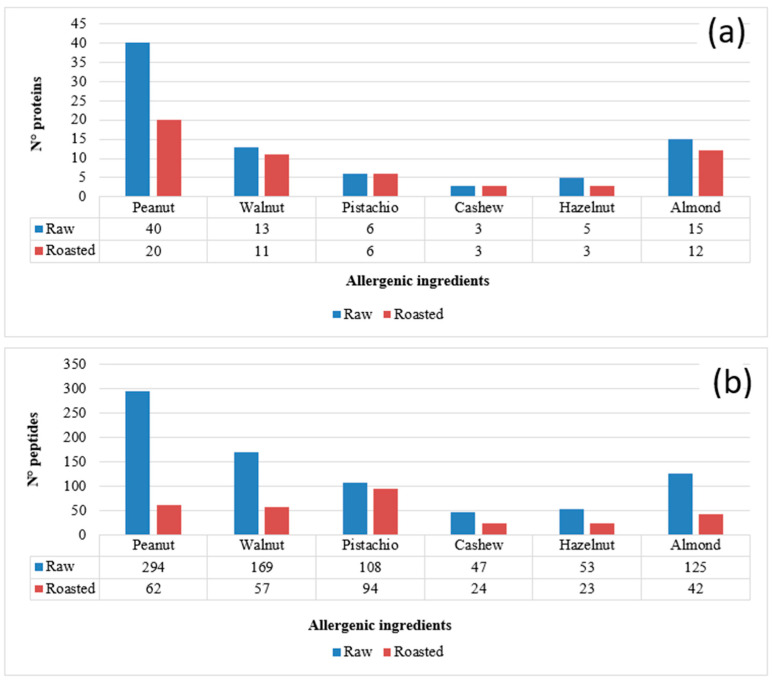
Numbers of proteins (**a**) and peptides (**b**) identified after the LC-ESI-MS and MS/MS analysis of the tryptic digests of protein extracts obtained from the five tree nuts and peanuts under study, considered as raw or roasted product. Protein identification was based on the Proteome Discoverer™ software operating on tryptic peptide MS/MS spectra obtained through Data Dependent™ MS/MS acquisitions and against a customized food allergen database. Bioinformatic filters used: (i) number of peptides recognized for each protein ≥ 2; (ii) Sequest HT score ≥ 1.

**Figure 2 foods-12-00726-f002:**
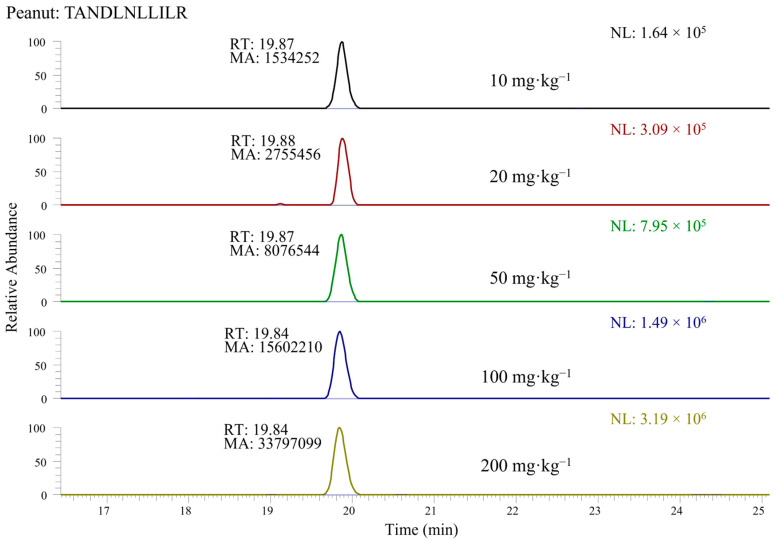
Example of XIC chromatograms related to the doubly charged peptide TANDLNLLILR (belonging to the peanut protein Ara 3) detected in the tryptic digests of protein extracts of cookies with concentrations of allergenic ingredients ranging from 10 to 200 mg·kg^−1^.

**Figure 3 foods-12-00726-f003:**
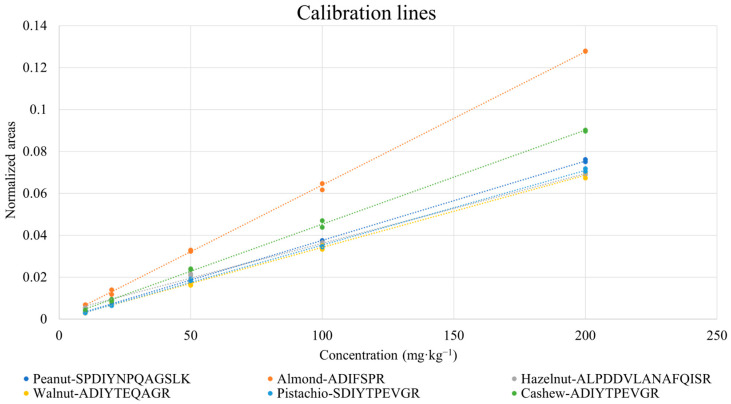
Examples of the calibration lines obtained for selected marker peptides of nuts and peanuts after the analysis of cookie protein extracts corresponding to different concentrations of allergenic ingredients.

**Table 1 foods-12-00726-t001:** Isotopically labelled peptides used as internal standards. The charge state (+2) and the theoretical *m*/*z* value of the ion used for each peptide are also reported.

Allergenic Ingredient	Protein	Peptide Sequence	Charge	*m*/*z* (Theoretical Values)
Peanut	Ara h 3—Cupin	SPDIYNPQAGSL(K)	+2	699.3612
		TANDLNLLIL(R)	+2	633.3764
Hazelnut	Cor a 9—11S Seed Storage Globulin	ADIYTEQVG(R)	+2	581.2925
		ALPDDVLANAFQIS(R)	+2	820.4373
Almond	Pru du 6—Amandin, 11S Globulin	TEENAFINTLAG(R)	+2	723.3666
		ADIFSP(R)	+2	408.2180

**Table 2 foods-12-00726-t002:** List of common peptide sequences detected in the tryptic digests of protein extracts obtained from raw or roasted tree nuts and peanuts and complying with constraints posed during this study (sequence length between 7 and 20, absence of C and M and of the NG motif, and absence of missed cleavages). The protein to which each peptide belongs is reported, along with its UniProt database accession number. The exact *m*/*z* ratio for the singly charged ion is also reported for each peptide. Underlined sequences correspond to peptides that were reliably identified in the tryptic digest of proteins extracted from an in-house prepared cookie incurred with 200 mg·kg^−1^ of each allergenic ingredient. Among them, sequences written also with bold characters were finally selected for the quantification of tree nuts and peanuts in the incurred cookies. See text for details.

Allergenic Ingredient	Peptide Sequence	Protein	Accessions UniProt Database	Exact *m/z* ratio for the Singly Charged Peptide
Peanut	ADFYNPAAGR	Legumin	A0A445AEY9	1081.5061
	ANLDAFTR	Desiccation-related protein PCC13-62-like	A0A445AWA2	907.4632
	EGEQEWGTPGSHVR	Ara h 1	N1NG13	1568.7088
	FFVPPFQQSPR	Arachin 6	A1DZF0	1349.7001
	FNLAGNHEQEFLR	Arachin 6	A1DZF0	1574.771
	**GTGNLELVAVR**	Ara h 1	B3IXL2	1128.6372
	LNALTPDNR	Legumin	A0A445AEY9	1013.5374
	NALFVPHYNTNAHSIIYALR	Arachin 6	A1DZF0	2314.2091
	NNPFYFPSR	Ara h 1	B3IXL2	1141.5425
	QILQNLR	Arachin 6	A0A444YLX0	884.5312
	QIVQNLR	Glycinin	O82580	870.5156
	SFNLDEGHALR	Ara h 1	B3IXL2	1258.6175
	**SPDIYNPQAGSLK**	Glycinin	O82580	1389.7009
	SSDNEGVIVK	Ara h 1	B3IXL2	1047.5317
	SSNPDIYNPQAGSLR	Ara h 3	E5G077	1618.782
	**TANDLNLLILR**	Glycinin	O82580	1255.7369
	TPQEILR	Desiccation-related protein PCC13-62-like	A0A445AWA2	856.4887
	VGDVFFVPR	Cupin	A0A445AA44	1035.5622
	VPGGFFPR	Desiccation-related protein PCC13-62-like	A0A445EM48	876.4727
	WGPAEPR	Ara h 1	B3IXL2	812.405
	WGPAGPR	Ara h 1	N1NG13	740.3838
	WLGLSAEYGNLYR	Arachin 6	A1DZF0	1541.7747
Walnut	ADIYTEEAGR	Jug r 4	Q2TPW5	1124.5218
	**ADIYTEQAGR**	11S globulin-like	A0A2I4F6R4	1123.5378
	AIPEEVLANAFQIPR	11S globulin-like	A0A2I4F6R4	1667.9115
	ALPEDVLINAYR	legumin B-like	A0A2I4GEH1	1373.7423
	ALPEEVLATAFQIPR	Jug r 4	Q2TPW5	1654.9163
	ATLTLVSQETR	Jug r 2.0101	Q9SEW4	1218.6688
	EGDVFAVPR	vicilin-like seed storage protein	A0A2I4F3W3	989.5051
	ELAFNFPAR	Jug r 6.0101	A0A2I4E5L6	1064.5524
	FYLAGNPHQQQQGGR	legumin B-like	A0A2I4GEH1	1700.8252
	GIIVTVEDELR	legumin B-like	A0A2I4GEH1	1243.6892
	HNLDTQTESDVFSR	Jug r 6.0101	A0A2I4EG83	1648.7562
	INALAGR	legumin B-like	A0A2I4GEH1	714.4257
	**INNLNAQEPGR**	legumin B-like	A0A2I4GEH1	1225.6284
	ISTVNSQNLPILR	11S globulin-like	A0A2I4F6R4	1454.8326
	ITSLNSFNLPILR	legumin B-like	A0A2I4GEH1	1487.858
	LDALEPTNR	Jug r 4	Q2TPW5	1028.5371
	LVALEPSNR	11S globulin-like	A0A2I4F6R4	998.5629
	LVYVVQGR	legumin B-like	A0A2I4GEH1	933.5516
	NEGFEWVSFK	Jug r 4	-Q2TPW5	1242.579
	NNIVNEFEK	Jug r 6.0101	A0A2I4E5L6	1106.5477
	QETFLAR	11S globulin-like	A0A2I4F6R4	864.4574
	**SFFLAGGEPR**	11S globulin seed storage protein 2-like	A0A2I4F669	1080.5473
	SFLLAGGEPR	Jug r 6.0101	A0A2I4EG83	1046.5629
	SGPSYQQIR	Jug r 6.0101	A0A2I4E5L6	1035.5218
	SPDQSYLR	Jug r 2.0101	Q9SEW4	965.4687
	SSGGPISLK	Jug r 2.0101	Q9SEW4	845.4727
	VFSNDILVAALNTPR	Jug r 2.0101	Q9SEW4	1629.8959
	WLQLSAER	Jug r 4	Q2TPW5	1002.5367
	YIQLSAER	legumin B-like	A0A2I4GEH1	979.5207
Pistachio	ADVYNPR	Pis v 2.0101	B7P073	834.4104
	ALPLDVIK	Pis v 2.0101	B7P073	868.5502
	DTDILAAFR	Ribulose bisphosphate carboxylase large	A0A1 × 9ZER6	1021.5313
	**EDAWNLK**	Pis v 2.0101	B7P074	875.4258
	EVLEAALK	Pis v 3.0101	B4 × 640	872.5088
	FEWISFK	Pis v 5.0101	B7SLJ1	956.4876
	FEWVSFK	Pis v 2.0101	B7P073	942.472
	FLQLSAK	Pis v 2.0101	B7P073	806.4771
	FLQLSVEK	Pis v 2.0101	B7P074	963.551
	FVLGGSPQQEIQGSGQSR	Pis v 2.0101	B7P073	1874.9355
	GDLQVIRPPR	Pis v 5.0101	B7SLJ1	1150.6691
	GFESEEESEYER	Pis v 5.0101	B7SLJ1	1490.59177
	GLPLDVIQNSFDISR	Pis v 2.0101	B7P074	1673.8857
	IAIVVSGEGR	Pis v 3.0101	B4 × 640	1000.5786
	ILAEVFQVEQSLVK	Pis v 5.0101	B7SLJ1	1602.9101
	IPTAYTK	Ribulose bisphosphate carboxylase large	A0A1 × 9ZER6	793.4454
	ISQLAGR	Pis v 2.0101	B7P074	744.4363
	ITSLNSLNLPILK	Pis v 5.0101	B7SLJ1	1425.8675
	LNINDPSR	Pis v 2.0101	B7P074	928.4847
	LQELYETASELPR	Pis v 1	B7P072	1548.7904
	**SDIYTPEVGR**	Pis v 5.0101	B7SLJ1	1136.5582
	**SETTIFAPGSSSQR**	Pis v 2.0101	B7P073	1467.7074
	STGTFNLFK	Pis v 3.0101	B4 × 640	1142.6204
	TFQGPPHGIQVER	Ribulose bisphosphate carboxylase large	A0A1 × 9ZER6	1465.7546
	VQEDLEVLSPHR	Pis v 2.0101	B7P073	1421.7383
	VTSINALNLPILR	Pis v 2.0101	B7P074	1423.8631
	WLQLSAER	Pis v 5.0101	B7SLJ1	1002.5367
	YNINDPSR	Pis v 2.0101	B7P073	978.4639
Cashew	**ADIYTPEVGR**	Ana o 2.0101	Q8GZP6	1120.5633
	DVFQQQQQHQSR	Ana o 2.0101	Q8GZP6	1528.7251
	**ELYETASELPR**	2s albumin	Q8H2B8	1307.6478
	FEWISFK	Ana o 2.0101	Q8GZP6	956.4876
	FHLAGNPK	Ana o 2.0101	Q8GZP6	883.4785
	IDYPPLEK	Vicilin-like protein	Q8L5L6	974.5193
	**LTTLNSLNLPILK**	Ana o 2.0101	Q8GZP6	1439.8832
	VFDGEVR	Ana o 2.0101	Q8GZP6	821.4152
	WLQLSVEK	Ana o 2.0101	Q8GZP6	1002.5619
	YGQLFEAER	Vicilin-like protein	Q8L5L6	1112.5371
Hazelnut	**ADIYTEQVGR**	11S globulin-like protein	A0A0A0P7E3	1151.5691
	**ALPDDVLANAFQISR**	11S globulin-like protein	A0A0A0P7E3	1629.8595
	ALSQHEEGPPR	Cor a 11.0101	Q8S4P9	1220.6018
	ELAFNLPSR	Cor a 11.0101	Q8S4P9	1046.5629
	GNIVNEFER	Cor a 11.0101	Q8S4P9	1077.5324
	HFYLAGNPDDEHQR	11S globulin-like protein	A0A0A0P7E3	1698.7619
	**INTVNSNTLPVLR**	11S globulin-like protein	A0A0A0P7E3	1440.8169
	IWPFGGESSGPINLLHK	Cor a 11.0101	Q8S4P9	1851.9752
	LLSGIENFR	Cor a 11.0101	Q8S4P9	1048.5786
	NIVKVEGR	11S globulin-like protein	A0A0A0P7E3	914.5418
	QGQQQFGQR	11S globulin-like protein	A0A0A0P7E3	1076.5232
	VQVLENFTK	Cor a 11.0101	Q8S4P9	1077.5939
	WLQLSAER	11S globulin-like protein	A0A0A0P7E3	1002.5367
Almond	ADFYNPQGGR	Prunin 2	A0A5E4FK23	1124.512
	**ADIFSPR**	Pru du 6.0101	A0A5E4FFS0	805.4203
	ALPDEVLANAYQISR	Pru du 6.0101	A0A5E4FFS0	1659.8701
	ALPDEVLQNAFR	Prunin 2	A0A5E4FK23	1372.7219
	ELAFNVPAR	Vicilin	A0A5E4EE27	1016.5524
	FEEFFPAGSR	Vicilin	A0A5E4EZP4	1186.5528
	FVSEDGIDNVR	(R)-mandelonitrile lyase	A0A5E4GEN6	1250.6012
	FYEASPQEFK	Vicilin	A0A5E4F2T7	1245.5786
	GNLDFVQPPR	Pru du 6.0101	A0A5E4FFS0	1142.5953
	LGFSSSLLFR	Gamma conglutin 1	P82952	1126.6255
	LKENIGNPER	Pru du 6.0101	A0A5E4FFS0	1169.6273
	**LLSATSPPR**	Prunin 2	A0A5E4FK23	941.5415
	LSQNIGDPSR	Prunin 2	A0A5E4FK23	1086.5538
	NQIIQVR	Pru du 6.0101	A0A5E4FFS0	870.5156
	QAYPWWR	Vicilin	A0A5E4F2T7	1006.4894
	QSYFVPASR	Vicilin	A0A5E4F2T7	1054.5316
	SLIGLAGTTPDR	Non-specific lipid-transfer protein	A0A4Y1RRI6	1200.6583
	**TEENAFINTLAGR**	Pru du 6.0101	A0A5E4FFS0	1435.7176
	VQGQLDFVSPFSR	Prunin 2	A0A5E4FK23	1479.7591

**Table 3 foods-12-00726-t003:** Correspondence between marker peptides selected for nuts and peanuts and isotopically labelled peptides used for the normalization of XIC peak areas.

Allergenic Species	Marker Peptides		Labelled Peptides Used for Signal Normalization	
	Sequence	*m/z* (+2)	Sequence	*m/z* (+2)
Peanut	GTGNLELVAVR	564.8222	ADIYTEQVG(R)	581.2925
	SPDIYNPQAGSLK	695.3541	SPDIYNPQAGSL(K)	699.3612
	TANDLNLLILR	628.3721	TANDLNLLIL(R)	633.3764
Walnut	ADIYTEQAGR	562.2726	ADIYTEQVG(R)	581.2925
	INNLNAQEPGR	613.3178	TANDLNLLIL(R)	633.3764
	SFLLAGGEPR	523.7851	ADIYTEQVG(R)	581.2925
Pistachio	EDAWNLK	438.2165	ADIYTEQVG(R)	581.2925
	SDIYTPEVGR	568.7828	ADIYTEQVG(R)	581.2925
	SETTIFAPGSSSQR	734.3573	TEENAFINTLAG(R)	723.3666
Cashew	ADIYTPEVGR	560.7853	ADIYTEQVG(R)	581.2925
	ELYETASELPR	654.3275	ADIYTEQVG(R)	581.2925
	LTTLNSLNLPILK	720.4452	TEENAFINTLAG(R)	723.3666
Hazelnut	ADIYTEQVGR	576.2882	ADIYTEQVG(R)	581.2925
	ALPDDVLANAFQISR	815.4334	ALPDDVLANAFQIS(R)	820.4373
	INTVNSNTLPVLR	720.9121	TEENAFINTLAG(R)	723.3666
Almond	ADIFSPR	403.2138	ADIFSP(R)	408.218
	LLSATSPPR	471.2744	ADIFSP(R)	408.218
	TEENAFINTLAGR	718.3624	TEENAFINTLAG(R)	723.3666

**Table 4 foods-12-00726-t004:** Correlation coefficients and detection (LOD) and quantification limits (LOQ) values obtained after calibrations performed for the six allergenic ingredients of interest in the cookie matrix considering specific marker peptides.

Species	Peptides	R^2^	LOD (mg·kg^−1^) *	LOQ (mg·kg^−1^) *
Peanut	GTGNLELVAVR	0.9984	4.4	15
	SPDIYNPQAGSLK	0.9995	2.4	8
	TANDLNLLILR	0.9977	5.2	17
Walnut	ADIYTEQAGR	0.9987	3.9	13
	INNLNAQEPGR	0.9987	3.8	13
	SFLLAGGEPR	0.9969	6.1	20
Pistachio	EDAWNLK	0.9981	4.7	16
	SDIYTPEVGR	0.999	3.4	11
	SETTIFAPGSSSQR	0.9989	3.6	12
Cashew	ADIYTPEVGR	0.9991	3.2	11
	ELYETASELPR	0.9957	7.1	24
	LTTLNSLNLPILK	0.9945	8.1	27
Hazelnut	ADIYTEQVGR	0.9981	4.7	16
	ALPDDVLANAFQISR	0.9982	4.5	15
	INTVNSNTLPVLR	0.9987	3.9	13
Almond	ADIFSPR	0.9995	2.3	7.8
	LLSATSPPR	0.9838	14	47
	TEENAFINTLAGR	0.9996	2.3	7.6

* The LOD and LOQ values refer to mg of allergenic ingredient/kg of cookie matrix.

**Table 5 foods-12-00726-t005:** VITAL^®^ 3.0 reference doses (also reported in the document “VITAL 3.0′: New and updated proposals for reference doses of food allergens”, freely accessible on the Internet with DOI: 10.17590/20200602-143608) and action levels for the five nuts under study and for peanuts. Data are referred to ED01 and 50 g reference amount. Data source: https://vital.allergenbureau.net/dashboard/home/ (accessed on 5 December 2022).

Species	Reference Dose (mg Proteins)	Action Level 1 (mg·kg^−1^)	Action Level 2 (mg·kg^−1^)
Cashew	0.05	<1.0	≥1.0
Pistachio	0.05	<1.0	≥1.0
Almond	0.10	<2.0	≥2.0
Hazelnut	0.10	<2.0	≥2.0
Walnut	0.03	<0.6	≥0.6
Peanut	0.20	<4.0	≥4.0

**Table 6 foods-12-00726-t006:** LOQ and LOD values, expressed as mg of total proteins per kg of cookie (mg·kg^−1^), achieved with the method developed during this study. The values were obtained from LOD and LOQ values reported in Table 4 (mg of allergenic ingredient per kg of cookie) after considering data on the total protein content of each ingredient. For the sake of comparison, threshold levels inferred for each species from the VITAL^®^ program, based on the ED01 eliciting dose and referred to a 50 g portion, are reported in the last column. See text for details.

Species	Selected Peptides	Protein Content (%) *	LOD (mg·kg^−1^)	LOQ (mg·kg^−1^)	Thresholds (mg·kg^−1^) for PAL Labelling According to VITAL^®^ 3.0
Peanut	GTGNLELVAVR	25.2	1.1	3.7	
	SPDIYNPQAGSLK		0.6	2	4
	TANDLNLLILR		1.3	4.3	
Walnut	ADIYTEQAGR	15.9	0.6	2	
	INNLNAQEPGR		0.6	2	0.6
	SFFLAGGEPR		0.9	3.2	
Pistachio	EDAWNLK	23.8	1.1	3.7	
	SDIYTPEVGR		0.8	2.7	1
	SETTIFAPGSSSQR		0.8	2.9	
Cashew	ADIYTPEVGR	19	0.6	2	
	ELYETASELPR		1.3	4.5	1
	LTTLNSLNLPILK		1.5	5.1	
Hazelnut	ADIYTEQVGR	16.4	0.8	2.6	
	ALPDDVLANAFQISR		0.7	2.5	2
	INTVNSNTLPVLR		0.6	2.2	
Almond	ADIFSPR	21.7	0.5	1.7	
	LLSATSPPR		3.0	10	2
	TEENAFINTLAGR		0.5	1.6	

* Source: Dutch National Institute for Public Health and the Environment—Food Composition Database (https://nevo-online.rivm.nl/Home/En (accessed on 26 October 2022)).

**Table 7 foods-12-00726-t007:** Results of the evaluation of intra-day and inter-day precision for the developed method, based on the analysis of the tryptic digest of a protein extract corresponding to a cookie incurred with 50 mg·kg^−1^ of each ingredient. Five LC-MS analysis replicates were performed during the same day on the same protein extract, for three consecutive days. The reported percentual coefficients of variation (CV%) were inferred from XIC peak areas for the selected peptides.

Species	Selected Peptides	Intra-Day			Inter-Day
		CV%	CV%	CV%	CV%
		DAY 1	DAY 2	DAY 3	DAYS 1,2,3
Peanut	GTGNLELVAVR	7.8	2.8	8.7	6.2
	SPDIYNPQAGSLK	1.6	1.7	1.6	2.1
	TANDLNLLILR	3.2	3.7	3.0	5.5
Walnut	ADIYTEQAGR	2.4	2.3	0.8	3.0
	INNLNAQEPGR	5.2	3.5	1.0	4.9
	SFFLAGGEPR	9.6	4.5	1.3	7.1
Pistachio	EDAWNLK	3.6	6.4	7.1	4.6
	SDIYTPEVGR	3.0	1.9	3.9	2.5
	SETTIFAPGSSSQR	7.5	9.8	5.5	7.5
Cashew	ADIYTPEVGR	2.5	3	4.4	3.7
	ELYETASELPR	1.1	0.5	0.5	3.5
	LTTLNSLNLPILK	3.7	4.4	8.7	6.3
Hazelnut	ADIYTEQVGR	6.7	5.6	0.5	19
	ALPDDVLANAFQISR	6.4	3.2	8.7	6.1
	INTVNSNTLPVLR	5.6	1.8	3.1	4.1
Almond	ADIFSPR	3.7	2.6	1.6	3.3
	LLSATSPPR	2.1	1.8	3.5	2.6
	TEENAFINTLAGR	3.1	3.9	0.7	3.2

## Data Availability

Data are available upon request made to the Corresponding Author.

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
