# Peer review of "Multi-Target Detection of Nuts and Peanuts as Hidden Allergens in Bakery Products through Bottom-Up Proteomics and High-Resolution Mass Spectrometry"

_foods, 2023, doi:10.3390/foods12040726_

Round 1

Reviewer 1 Report

 The manuscript entitled " Multi-target detection of nuts and peanuts as hidden allergens in bakery products through bottom-up proteomics and high-3 resolution mass spectrometry” (ref foods-2175843) by Luparelly  et al. presents interesting results on the innovative development of a MS technique to the simultaneous and independent detection of nuts and peanut in processed food.

In my opinion, this manuscript is technically good, the experimental work is well designed and the results are relevant in the field of the detection of food allergens.

The fact to include several allergenic foods in a single assay is a great advantage as some nuts and peanut are often found together as hidden allergens in food and then, it would allow significant time and cost savings.

On the other hand, the lack of reference materials to validate analytical techniques for the detection of allergens makes important to work with model foods prepared with a known content of allergenic ingredients and that have been processed under conditions equivalent to those used in the food industry. This allows assessing the functionality of analytical techniques under real industrial processing conditions.

The method developed to determine nuts sand peanut shows a high sensitivity  to protect allergic consumers, considering data about protein action levels reported by the VITAL program.

However, I consider the manuscript needs some minor corrections.

Minor comments:

Introduction. Line 65. Cor a 9 should be included among allergenic proteins from hazelnut. This 11S globulin-like protein is  one of the more abundant and allergenic protein from hazelnut and its peptides have been also  selected by authors as a good marker for the presence of  that nut. At this respect, Pages 63-67. Authors should revise the WHI-IUIIS database about allergenic proteins of nuts

Introduction. Line 104. Lateral Flow Immuno Assay (LFIA) should be included as recently it is an immunochemical technique widely used in food industry to detect allergens due to its quickness and simplicity.  Furthermore, the hook effect is a problem associated only to LFIA and not to ELISA assays (line 106). This should be corrected in the manuscript. Besides, a paper has been recently published on the application of an LFIA technique to the independent determination of almond and peanut in one assay (Biosensors, 2022, 12(11), 980).

Methods. Line 191. Did authors perform some experiments to determine the homogeneity of the 1000 ppm sample prepared? .

Methods. Line 289. Is the flow rate of 200 mL/min correct?. It looks a very high flow. Maybe  units are wrongly indicated.

Methods. Line 335. Authors could have included some level of contamination lower than 10 ppm, as it seems that LOD is lower than that value (Table 4).

Line 355. Five LC-MS analysis replicates is referred to five independent extractions or they are the same extract analysed five times? It should be clarified in the manuscript.

Table 5. Data given correspond to reference dose (mg) as it is indicated in the first column of numbers. This information is given in the web document  (https://mobil.bfr.bund.de/cm/349/vital-30-new-and-updated-proposals-for-reference-doses-of-food-allergens.pdf) DOI 10.17590/20200602-143608 and thus,  it should be cited in the Table. However, data are not referred to a 50 g reference amount, as those data would be higher (as stated in Table 6). See reference by Holzhauser et al,2020, Food and Chemical Toxicology, 145, 111709.

Tables 4 and 5 could be combined and results given together to make them more easy to compare.

Table 7. Was the extract of a cookie sample analysed or extracts from five independent cookie samples?.

Author Response

We thank reviewer 1 for his/her comments issued on the manuscript. Please find hereby attached the point-to-point replies 

Reviewer 2 Report

GENERAL COMMENTS:
The authors describe in this manuscript a method for the detection and quantification of major allergens from the main nuts and peanut species using a bottom-up proteomics approach applied to a in-house preparation (cookie). The authors' scientific approach is well-detailed and the method developed is promising, particularly for the quantification of traces of nuts and peanuts in processed products and to look for contamination (before cooking) in bakery products. The results are well-structured and presented clearly, showing interesting detection and quantification limits, consistent with current standards (VITAL program) and relatively good precision and reproducibility. Overall, the article is of good quality, the methods and results are well-detailed and the writing is excellent. However, it would be beneficial to explore the potential application and validation of this method in real-world scenarios in order to assess its performance and feasibility in a broader context.

SPECIFIC RECOMMENDATIONS:
- lines 24-26 & methods/results sections: it would be interesting to consider and/or discuss the impact not only of cooking but also of protein denaturation induced by the digestion method used in the study, given that such alterations can also lead to the creation of antigenic neo-epitopes and/or change protein allergenicity.
- lines 27-30 & results section: it would be appropriate to further discuss the clinical implications and applications (results in ppm versus eliciting thresholds typically expressed in grams in clinical practice, re-evaluation of the meaning of "traces" in product labeling...)
- lines 61-67: insufficient reference literature (missing some clinically relevant major/minor molecular allergens)
- line 615: naming error - "Figure 7" --> "Table 7"
- lines 616-617: there are still exceptions to this conclusion, so it would be more fair to be less assertive and maybe to discuss the role of the specific characteristics of the peptides used as well as the methodology in this variability of results.
- lines 638-642: discuss the limit regarding the case of very low eliciting thresholds (high-risk patients)
- lines 645-647: the approach is promising, but is it scalable? It would seem relevant to discuss the limitations, particularly the costs and technical feasibility (operators, equipment, reagents).

Author Response

We thank reviewer 2 for his/her comments issued on the manuscript. Please find hereby attached the point-to-point replies 

Reviewer 3 Report

The manuscript entitled Multi-target detection of nuts and peanuts as hidden allergens in bakery products through bottom-up proteomics and high resolution mass spectrometry by Anna Luparelli is based on a novel idea and is interesting.

Before recommending its publication I would ask the authors to address the following comments,

As we know that  Food allergens have presented the food industry with a new paradigm for chemical food safety because small amounts of an allergenic food, innocuous to the vast majority of consumers, can pose a significant risk of eliciting a reaction in allergic consumers. There is currently no accepted cure, so individuals with food allergies have to practice food avoidance. To help them, regulations have been enacted across the world that require priority allergenic foods to be listed on ingredient labels at whatever their level of inclusion in a recipe. 

For detection of various bonds

Why the authors did not include metabolomics in their methodology?

Instead of Bioinformatic analysis wasn’t it better to extract the allergen part from the nuts?

Because bioinformatic/ in silico analysis will give you just predicted allergen.

The authors should extract the allergen part from the nut and then do its evaluation via chromatographic analysis i.e. GCMS LCMS HPLC etc

I would also suggest the authors to do FTIR and XRD analysis of the control and different samples.

The authors are advised to do SDS-PAGE Analysis of the Raw and roasted peanut extracts were by 1D PAGE.

There are some topographical mistakes and some grammatical mistakes which should be removed before submitting the revised manuscript.

Overall the scientific soundness and novelty of the manuscript is very good.

I would suggest minor revision.

Author Response

We thank reviewer 3 for his/her comments issued on the manuscript. Please find hereby attached the point-to-point replies 
